# Mechanistic insight into the chemical treatments of monolayer transition metal disulfides for photoluminescence enhancement

Zhaojun Li [1,2], Hope Bretscher [1], Yunwei Zhang [1], Géraud Delport [1], James Xiao[1], Alpha Lee [1], Samuel D. Stranks [1,3] & Akshay Rao [1✉]

There is a growing interest in obtaining high quality monolayer transition metal disulfides for optoelectronic applications. Surface treatments using a range of chemicals have proven effective to improve the photoluminescence yield of these materials. However, the underlying mechanism for the photoluminescence enhancement is not clear, which prevents a rational design of passivation strategies. Here, a simple and effective approach to significantly enhance the photoluminescence is demonstrated by using a family of cation donors, which we show to be much more effective than commonly used p-dopants. We develop a detailed mechanistic picture for the action of these cation donors and demonstrate that one of them, bis(trifluoromethane)sulfonimide lithium salt (Li-TFSI), enhances the photoluminescence of both $MoS_2$ and $WS_2$ to a level double that of the currently best performing super-acid trifluoromethanesulfonimide (H-TFSI) treatment. In addition, the ionic salts used in our treatments are compatible with greener solvents and are easier to handle than super-acids, providing the possibility of performing treatments during device fabrication. This work sets up rational selection rules for ionic chemicals to passivate transition metal disulfides and increases their potential in practical optoelectronic applications.

[1] Cavendish Laboratory, University of Cambridge, JJ Thomson Avenue, CB3 0HE Cambridge, UK. [2] Molecular and Condensed Matter Physics, Department of Physics and Astronomy, Uppsala University, 75120 Uppsala, Sweden. [3] Department of Chemical Engineering & Biotechnology, University of Cambridge, Philippa Fawcett Drive, CB3 0AS Cambridge, UK. ✉email: ar525@cam.ac.uk

The discovery of 2D materials based on semiconducting transition metal disulfides (TMDSs), with the chemical structure MS$_2$ (M = Mo, W), has opened up new interesting possibilities in optoelectronic devices, as monolayer TMDSs possess direct bandgaps with absorption in the visible spectral region, as well as other excellent properties well suited for optoelectronic applications, like high extinction coefficients due to the strong excitonic effects, exceptional mechanical properties, and chemical and thermal stability[1–4]. Nevertheless, monolayer TMDSs often exhibit poor photoluminescence quantum yields (PLQYs), which is the key figure of merit for optoelectronic devices[5]. Atomic vacancies, such as sulfur vacancies, which lead to trapping and non-radiative decay are thought to be the primary defects in these materials[6,7]. In addition, trion formation, which occurs easily in these materials, leads to non-radiative recombination and quenched photoluminescence (PL)[8–10]. This is especially problematic since as-prepared TMDSs are often doped[11,12].

To overcome these problems, there has been a large effort to develop chemical passivation strategies for 2D TMDSs[13–15]. Chemical passivation by completing the dangling bonds has been widely used in silicon solar cells to improve the device performance[16,17]. For TMDS materials, the controlled physisorption of small molecules on the TMDS surface is reported to be a viable approach to tune their optical and electronic properties, but the increase in PLQY with these treatments is modest[11,18,19]. In contrast, the use of 'acid treatment' with the super-acid trifluoromethanesulfonimide (H-TFSI) has been shown to greatly improve PLQY, up to 200 fold[20]. Despite these known treatments, the search for new passivating chemical treatments continues. The harsh nature of the H-TFSI super-acid limits its application in optoelectronic devices, where it can cause damage to both the TMDS material and contacts. Most importantly, the underlying mechanism of how these chemical treatments work is unclear. A reduction of n-doping and trion formation leading to increases in radiative recombination has been suggested as the mechanism of action, but this picture is clearly incomplete and hinders the rational design of new passivation schemes[9,21–24].

Here, by systematically studying the PL enhancement of TMDSs caused by different ionic chemicals, as well as widely used small molecule p-dopants, we are able to move beyond the simple picture of p-doping and provide mechanistic insight into the roles played by both the cation and counter anion during chemical treatments. We show that the strong PL improvement is caused by cation adsorption on the TMDS surface instead of simply charge transfer to molecular p-dopants and that the counter anion should be non-coordinating with strong electron-withdrawing groups. This allows us to introduce mild chemical treatments that use ionic salts compatible with a diverse range of green solvents, performed under ambient conditions. We demonstrate that bis(trifluoromethane)sulfonimide lithium salt (Li-TFSI) treatment yields PL improvements twice as large as H-TFSI treatment, as well as greatly improved exciton diffusion compared to pristine or H-TFSI treated samples.

## Results and discussion

The structures of all the chemical treatment agents used in this study are illustrated in Fig. 1a. The chemical treatments were achieved by immersing the monolayers in concentrated solutions of the investigated chemicals (0.02 M) for 40 min. Because the PL of MoS$_2$ increases while increasing the concentration of H-TFSI (Supplementary Fig. 1), we compare all treatments with a fixed concentration of 0.02 M. Figure 1b demonstrates the general PL enhancements on WS$_2$ with different chemical treatments. Apart

from the H-TFSI super-acid reported previously, we find that a range of TFSI based ionic salts lead to PL enhancements to varying degrees. Interestingly, calcium (II) bis(trifluoromethanesulfonimide) (Ca(TFSI)$_2$) and Li-TFSI greatly improve PL, beyond what can be achieved via H-TFSI and other chemical treatments. It is also worth noting that compared to H-TFSI, which needs to be dissolved in dichloroethane (DCE) and has to be handled in the glovebox due to being extremely hygroscopic, ionic salts such as Li-TFSI and Ca(TFSI)$_2$ function in various milder solvents like acetonitrile, isopropanol, and methanol, and can be easily handled in ambient atmosphere.

In the discussion to follow, we will focus on Li-TFSI, as an example of a TFSI anion based ionic salt, as Li-TFSI results in the highest PL enhancements in both MoS$_2$ and WS$_2$ (see discussion on the other ionic salts M$_1$-TFSI (M$_1$ = Na and K) and M$_2$(TFSI)$_2$ (M$_2$ = Mg, Ca and Cu) in SI). We start by comparing the improvement in PL intensity via treatment with H-TFSI and Li-TFSI. Representative PL spectra for pristine, H-TFSI, and Li-TFSI treated monolayers MoS$_2$ and WS$_2$ are shown in Fig. 1c and d[25]. The PL of pristine MoS$_2$ and WS$_2$ has large contributions from trions with peak emission at 663 and 626 nm, respectively (see SI for spectral deconvolution). After chemical treatments, the PL is greatly enhanced and the peak position blueshifts for both MoS$_2$ and WS$_2$ due to the suppression of trions[26]. In addition, the PL enhancement yielded by the Li-TFSI treatment is almost double that of the H-TFSI treatment, for both MoS$_2$ and WS$_2$ with an exciton emission peak at 659 nm and 617 nm, respectively. The blueshift in peak position is accompanied by a more uniform emission profile in both Li-TFSI and H-TFSI treated MoS$_2$ and WS$_2$. The distribution of the peak emission wavelength narrows with these treatments, as shown in scatter plots of the peak PL counts versus emission peak position acquired from PL spatial maps (Supplementary Fig. 2). Due to the PL inhomogeneity of TMDS samples, we conducted PL measurements on multiple WS$_2$ monolayer samples to obtain a better statistical picture of the enhancement factor (Supplementary Fig. 3, see discussion in SI). Other chemical treatments outlined in Fig. 1 exhibit various PL increase and peak position blueshift shown in Supplementary Figs. 4–10. The macroscopic effect of all chemical treatments is p doping. However, this simple picture of a doping-based chemical treatment mechanism cannot explain the widely varying magnitudes of the PL intensity change of different chemicals, as summarized in Fig. 1b. For example, H-TFSI is also much better than other acids in regard to the PL enhancement effect on TMDSs even though the p-doping H$^+$ ion is the same (Supplementary Fig. 8). The underlying question is thus why some of these treatments show much greater efficiency than others and what the roles of the anion and cation in the treatment process is.

To study the mechanism of the PL enhancement, we carried out Raman spectroscopy on pristine H-TFSI, and Li-TFSI treated monolayer MoS$_2$ samples. A multi-peak Lorentzian fitting is performed on each spectrum to extract the chemical treatment-dependent shift of the MoS$_2$ Raman peaks, as shown in Fig. 2. The in-plane $E_{2g}^1$ mode at 388 cm$^{-1}$ of pristine MoS$_2$ is associated with opposite vibration of two S atoms with respect to the Mo atom while the $A_{1g}$ Raman mode at 405 cm$^{-1}$ of pristine MoS$_2$ results from out-of-plane vibration of S atoms in opposite directions and is sensitive to doping-induced electron density[27]. The second-order Raman resonance 2LA mode at 442 cm$^{-1}$ involving longitudinal acoustic phonons is assigned to in-plane collective movements of the atoms in the lattice[28]. After the treatment, both $A_{1g}$ and 2LA modes are blueshifted whereas the $E_{2g}^1$ mode is not affected. This is attributed to the weaker electron-phonon coupling caused by the adsorption of the cations. Interestingly, a new Raman mode at ~ 468 cm$^{-1}$ emerges after H-TFSI and Li-TFSI treatments. This is assigned to the $A_{2u}$

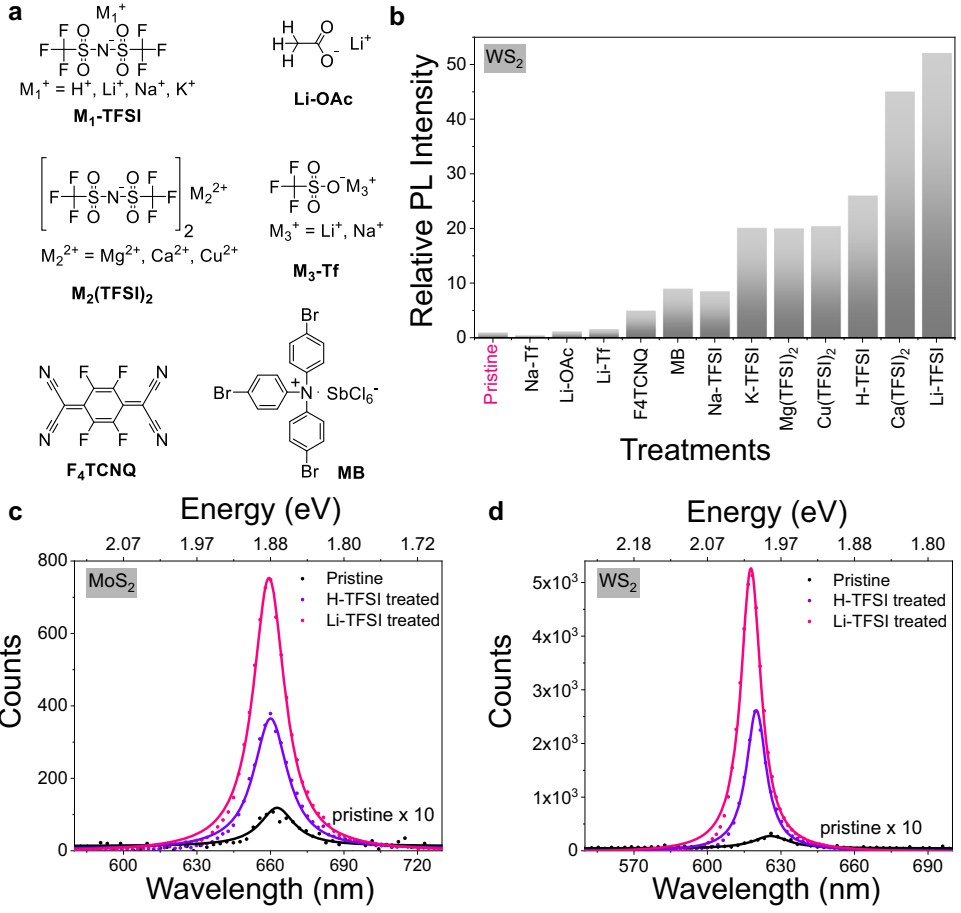

**Fig. 1 Studied chemicals and their steady-state photoluminescence (PL) enhancement on monolayers MoS₂ and WS₂. a** Structures of all the chemicals for the treatments. **b** General illustration of PL intensity enhancements on WS₂ with different chemical treatments compared to pristine sample (The PL intensity of pristine sample is normalized to 1). **c** Representative PL spectra for pristine, H-TFSI, and Li-TFSI treated monolayer MoS₂. The PL intensity of pristine MoS₂ is magnified 10 times for clarity. **d** Representative PL spectra for pristine, H-TFSI and Li-TFSI treated monolayer WS₂. The PL intensity of pristine WS₂ is magnified 10 times for clarity.

mode, which is Raman-silent due to the reflection symmetry in pristine $MoS_2$[29]. The results imply that $H^+$ and $Li^+$ ions can be adsorbed on the surface of $MoS_2$, perturbing the crystal lattice and activating the previously silent $A_{2u}$ mode. As the PL enhancement is significantly greater for H-TFSI and Li-TFSI than p-doping small molecules (see SI Supplementary Fig. 11), it suggests that PL modulation strength of the ionic chemicals might be determined by the interaction between the cation and the TMDS surface. This is in contrast to the common assumption that PL enhancement by surface chemical treatment is solely due to molecular p-dopant induced electron transfer[9,30].

To test our hypothesis of the importance of cations, the PL enhancements of $MoS_2$ and $WS_2$ treated with two common molecular p-dopants tris(4-bromophenyl)ammoniumyl hexa-chloroantimonate ("Magic Blue," MB) and 2,3,5,6-Tetra-fluoro-7,7,8,8-tetracyanoquinodimethane (F4TCNQ) were investigated[11,19,31]. As depicted in Supplementary Fig. 9, while both MB and F4TCNQ increased the PL of $MoS_2$ slightly, the enhancement is negligible in contrast to $M_1$-TFSI ($M_1$ = H, Li, Na, and K) and $M_2$(TFSI)$_2$ ($M_2$ = Mg, Ca, and Cu) treatments. Similar small PL enhancement for MB and F4TCNQ-treated $WS_2$ are also seen (See Supple, Supplementary Fig. 10, for detailed discussion). Moreover, there is a clear trion con-tribution from the emission of MB-treated $MoS_2$, and the PL of F4TCNQ-treated $MoS_2$ is too weak to obtain an accurate fitting. As illustrated in Supplementary Fig. 11, both $A_{1g}$ and

2LA Raman modes of MB-treated and F4TCNQ-treated $MoS_2$ are slightly blueshifted due to the p-doping effect[31]. However, the shift is smaller compared to Li-TFSI-treated $MoS_2$, and there is no appearance of the $A_{2u}$ mode. This comparison strongly supports our hypothesis that PL enhancement of chemical-treated TMDSs is attributed to stable cation adsorption instead of electron transfer induced by molecular p-doping. This stable cation adsorption effectively suppresses trion formation in these materials.

Surface-sensitive X-ray photoelectron spectroscopy (XPS) measurements were also carried out on pristine, H-TFSI-treated, and Li-TFSI-treated $MoS_2$ samples to investigate the chemical treatment mechanism. As depicted in Supplementary Fig. 12, the F 1$s$ and Li 1$s$ core levels show clear signatures of Li-TFSI adsorption on the sample surface[32]. Moreover, there is no observable change in oxidation state or bonding property according to the Mo 3$d$ core levels which represent the Mo(IV) species[33-35]. Thus the peak at 169 eV in S 2$p$ core levels is assigned to the TFSI anion instead of new oxidation state for-mation during the Li-TFSI treatment[32]. Though the S 2$s$ peak, S 2$p$ doublet peaks, and Mo 3$d$ doublet peaks appear to shift towards higher oxidation states after H-TFSI and Li-TFSI treat-ments, the low sensitivity of the instrument prevents any inter-pretation of these effects. These results suggest that new bonds are not being formed by the treatments on the surface of the TMDSs. As we discussed in detail in the SI, results from washing of the

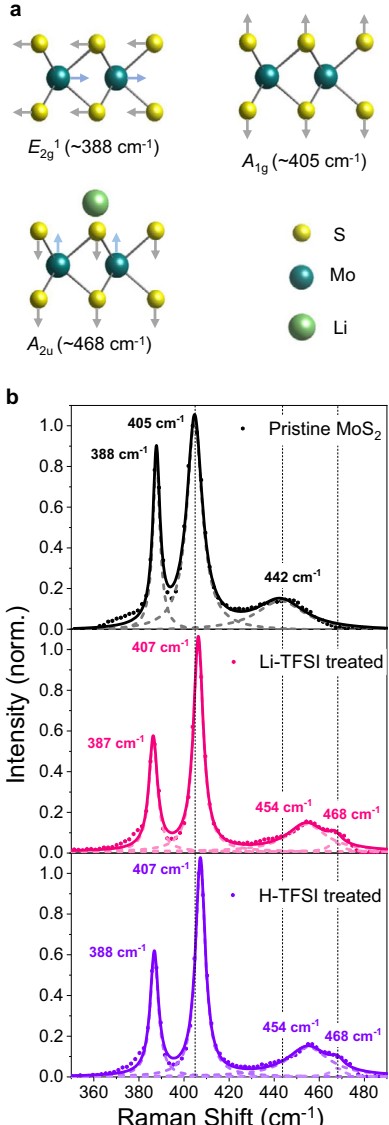

**Fig. 2 Raman spectra of chemically treated MoS₂ monolayers. a** Side views of Raman modes. **b** Raman spectra of pristine, H-TFSI-treated, and Li-TFSI-treated monolayer MoS₂. The decomposed Lorentzian peak fitting of each spectrum is presented as a short, dashed line and the cumulative fitting is presented as a solid line. The positions of $A_{1g}$ and 2LA mode of pristine MoS₂ as well as $A_{2u}$ mode of MoS₂ with adatom (Li for example) are illustrated in each spectrum with a short black dashed line for direct comparison. The value of each peak position is also stated in the spectra.

samples to remove the treatments also suggest that chemical reactions are not playing a role in the improvement of the PL. We are also able to rule out ion intercalation as has been suggested by previous studies (see Supplementary Note 3, page 9-11, Supplementary Figs. 7 and 8)[36].

To further test the above hypothesis of cation adsorption on the TMDSs, we investigate the stability of H and Li atom adsorbed at various types of adsorption sites in monolayer MoS₂ and WS₂ via density functional theory (DFT) simulations of the formation energy[37]. The formation energies of adatoms at sulfur vacancy sites ($E^{Sv}$), on top of sulfur ($E^{sf}(S)$), and on top of molybdenum ($E^{sf}(Mo)$) of MoS₂ are summarized in Table 1. The corresponding formation energy of cation-adsorbed WS₂, as well as the bond energy between cations and TFSI anion is listed in

Table S1 (see SI for detail). The calculated results show that adsorption on both sulfur vacancy sites and on the top of surfaces of MoS₂ are thermodynamically stable with negative formation energies, but that the sulfur vacancy site rather than the surfaces of TMDSs is the most favorable adsorption location for all adatoms. In general, the adsorptions of Li adatom are energetically more favorable at surface sites ($E^{sf}$) compared to H adatom. However, H adsorption energy at sulfur vacancy site ($E^{Sv}_H$) is slightly more stable than Li ($E^{Sv}_{Li}$). Considering that the material has more available adsorption sites at the surface than sulfur vacancies, we believe that with chemical treatments, the concentration of Li adatoms on MoS₂ is higher than that of H adatom, due to the availability of locations for adsorption. This supports our assumption that the trion formation will be strongly suppressed by a higher adsorption of cation, leading to superior PL enhancement of TMDSs with the Li-TFSI treatment.

To explore the photophysics after chemical treatments, we conducted time-resolved PL (TRPL) and ultrafast pump-probe measurements. The average lifetime versus PL intensity for different spots measured from a 2D map taken on H-TFSI-treated and Li-TFSI-treated monolayer MoS₂ samples are shown in Supplementary Fig. 13, which statistically illustrates that the radiative lifetime is strongly correlated with the PL intensity. The lifetimes shorten while the PL intensity enhances, suggesting that the radiative recombination rate increases upon chemical treatment. Normalized average TRPL decays of H-TFSI-treated and Li-TFSI-treated MoS₂ samples at room temperature show noticeably different exciton decay dynamics (Fig. 3a). The TRPL curves are fitted by a three-exponential decay function with average lifetime (<τ>) ~320 ps, and ~150 ps for H-TSFI-treated and Li-TFSI-treated MoS₂, respectively (Fitting results are shown in Table S2). The TRPL decay of pristine MoS₂ is not presented as it is below the instrument response function (IRF) limit (~100 ps). At room temperature, the decay components can be attributed to a variety of sources[38]. The longer lifetime upon H-TFSI treatment results from a trap-mediated exciton recombination process, which has been discussed in detail in previous studies and is supported by the following ultrafast pump-probe measurements[39]. In contrast, the shorter PL lifetime in Li-TFSI-treated MoS₂ indicates a greatly reduced role of exciton traps and is again consistent with the pump-probe results to follow.

The pump-probe spectra of pristine, H-TFSI-treated, and Li-TFSI-treated MoS₂ are depicted in Supplementary Fig. 14, Fig. 3b and c, respectively. The exciton dynamics of pristine and H-TFSI-treated MoS₂ samples have been discussed in detail in our previous study[40]. The principle ground state bleach features correspond to the A and B excitons, at around 660 and 600 nm respectively[41]. The exciton lifetime is lengthened in the H-TFSI-treated sample due to the repopulation of the A exciton via thermal activation out of trap sites related to sulfur vacancies. These sub-gap trap sites (sulfur vacancies), which appear as a positive feature at 730 nm in the pump-probe (ΔT/T) spectra, have been previously detected in H-TFSI-treated MoS₂ and been shown to lead to trap limited emission lifetimes. In contrast, no sub-gap defect state emerges in the pump-probe spectra of the Li-TFSI-treated MoS₂ sample (no photo-induced features seen at 730 nm), indicating the lack of sub-gap trap sites which then leads to a shorter exciton lifetime. The pump-probe results agree well with TRPL data where Li-TFSI-treated MoS₂ sample presents a shorter lifetime due to a lack of exciton trapping and suggests that excitons in Li-TFSI-treated MoS₂ recombine more efficiently bypassing trap states or that the subgap state formed due to sulfur vacancies is passivated.

Exciton transport is an important criterion in many optoelectronic devices and one that can be strongly affected by semiconductor properties such as doping and traps. Here, we directly

**Table 1 DFT simulation of H and Li adatoms formation energies and the configurations on the different positions of monolayer MoS₂.**

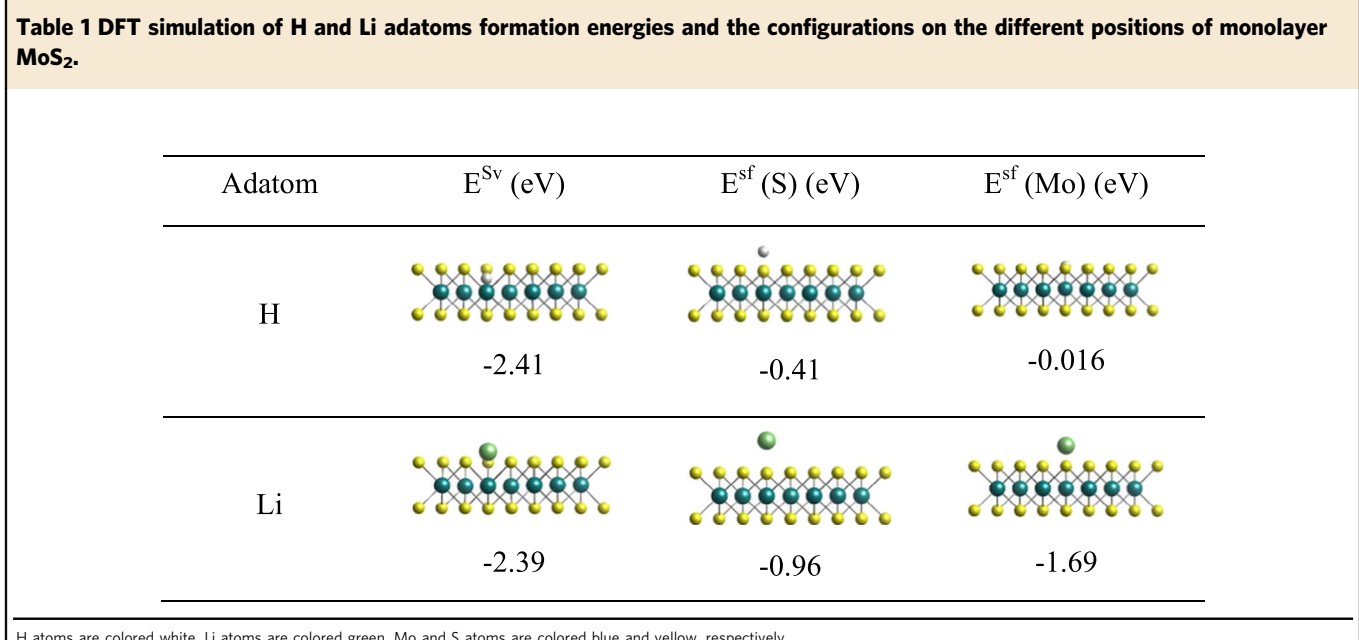

| Adatom | $E^{Sv}$ (eV) | $E^{sf}$ (S) (eV) | $E^{sf}$ (Mo) (eV) |
|---|---|---|---|
| H | -2.41 | -0.41 | -0.016 |
| Li | -2.39 | -0.96 | -1.69 |

H atoms are colored white. Li atoms are colored green. Mo and S atoms are colored blue and yellow, respectively.

monitor the spatial propagation of photogenerated excitons in H-TFSI-treated and Li-TFSI-treated MoS₂ monolayers on quartz substrates under ambient conditions with a confocal PL set up, as shown in Fig. 3b and c[42–44]. The Gaussian pump beam creates a Gaussian initial distribution population of excitons $n(x, 0)$ created by at position $(x_0)$, which is given by[45]

$$n(x, 0) = N \exp\left[-\frac{(x - x_0)^2}{2\sigma_0^2}\right] \quad (1)$$

with a variance of $\sigma_0^2$. In the following, the exciton density at any delay time $(t)$ will be approximated with another Gaussian function:

$$n(x, t) = N \exp\left[-\frac{(x - x_0)^2}{2\sigma_t^2}\right] \quad (2)$$

with a variance of $\sigma_t^2$. The normalized PL intensity profile ($I_{PL}$) at each time snapshot $(t)$ for H-TFSI-treated and Li-TFSI-treated monolayer MoS₂ are shown in Fig. 3d and e, respectively, together with the instrument response. For any considered time, the normalized PL profiles is well fitted with the Gaussian model. This allows us to extract the time evolution of the variance $\sigma_t^2$ for the two samples. At early time, $(t < 1\,\text{ns})$, $\sigma_t^2$ grows linearly with time, which is indicative of a diffusive motion of excitons[46]. At longer time, the value of $\sigma_t^2$ tends to saturate or even decrease for H-TFSI-treated MoS₂ sample. This indicates that the majority of propagating excitons has already decayed and that remaining ones are located around the point of creation $(x = 0)$. From the diffusive part of the curve, the exciton diffusion coefficient $(D)$ is extracted from the slope of the fitting lines (Fig. 3f), using the diffusion equation:

$$D = \frac{\sigma_t^2 - \sigma_0^2}{2t} \quad (3)$$

The higher $D_{\text{Li-TFSI}}$ value of 0.22 cm² s⁻¹ in the Li-TFSI-treated MoS₂ sample compared to $D_{\text{H-TFSI}}$ value of 0.1 cm² s⁻¹ in H-TFSI-treated MoS₂ sample indicates that excitons in Li-TFSI-treated MoS₂ sample propagate more efficiently without trapping.

In order to comprehensively understand the treatment mechanism, we also compare the PL intensity enhancements of TMDSs treated with Li⁺ and Na⁺ salts of different counter anions. Lithium triflate (Li-Tf) and sodium triflate (Na-Tf) were

employed for comparison in this work since the Tf anion shows great similarity to the TFSI anion and dissociates freely in solution. Lithium acetate (Li-OAc) was also selected to further explore the effect of the counter anions on PL modulation of TMDSs. The scatter plots of emission peak position and peak PL counts from PL maps of treated monolayer MoS₂ and WS₂ are shown in Supplementary Fig. 15. The PL of Na-Tf-treated and Li-OAc-treated MoS₂ showed no observable PL enhancement, hence these data are not presented. Representative PL spectra for Li-Tf-treated monolayers MoS₂ and WS₂ are shown in Fig. 4. Li-Tf treatment presents a clear PL enhancement for both MoS₂ and WS₂, whereas Li-OAc treatment only increases PL of WS₂ sample slightly. However, the improvement factors for both Li-Tf and Li-OAc on WS₂ are quite small compared to Li-TFSI treatment. The effect of Li-OAc on MoS₂ PL enhancement factor is difficult to determine since the PL of both pristine and Li-OAc-treated MoS₂ were unmeasurable. Moreover, there is clear trion emission contribution in the Li-Tf-treated MoS₂ at 664 nm. The results clearly suggest that counter anions play an important role in modulating the PL of TMDSs. The DFT simulations of Tf and TFSI anion adsorption at the sulfur vacancy sites of monolayer MoS₂ show that Tf anion tends to fill in the sulfur vacancy whereas there is no interaction between TFSI anions and the MoS₂ surface (Supplementary Fig. 16).

Based on the results presented here, we speculate there are two reasons why TFSI based ionic salts work so well to enhance the PL of TMDSs. The presence of two strong electron-withdrawing groups (-CF₃SO₂) on the same nitrogen atom leads to a significantly lower surface charge density of the TFSI anion compared to the Tf anion[47]. In addition, the bulky side groups (-CF₃SO₂) lead to huge steric hindrance and make TFSI non-coordinating, while the Tf anion can coordinate to Mo or W at surfaces of TMDSs and leads to trion formation behaving as a n-doping reagent. Therefore, there is a competition between Li⁺ and Tf anion adsorption on the surface of TMDS monolayer during the Li-Tf treatment. In the case of Na-Tf treatment of WS₂, the PL decreased due to the negligible effect of the Na⁺ cation and negative n-doping effect of the Tf anion. The weak effect of Li-OAc is, on the other hand, explained by the weak dissociation of ions. As illustrated in Supplementary Fig. 17a, the $A_{1g}$ and 2LA Raman modes for Li-Tf-treated MoS₂ are

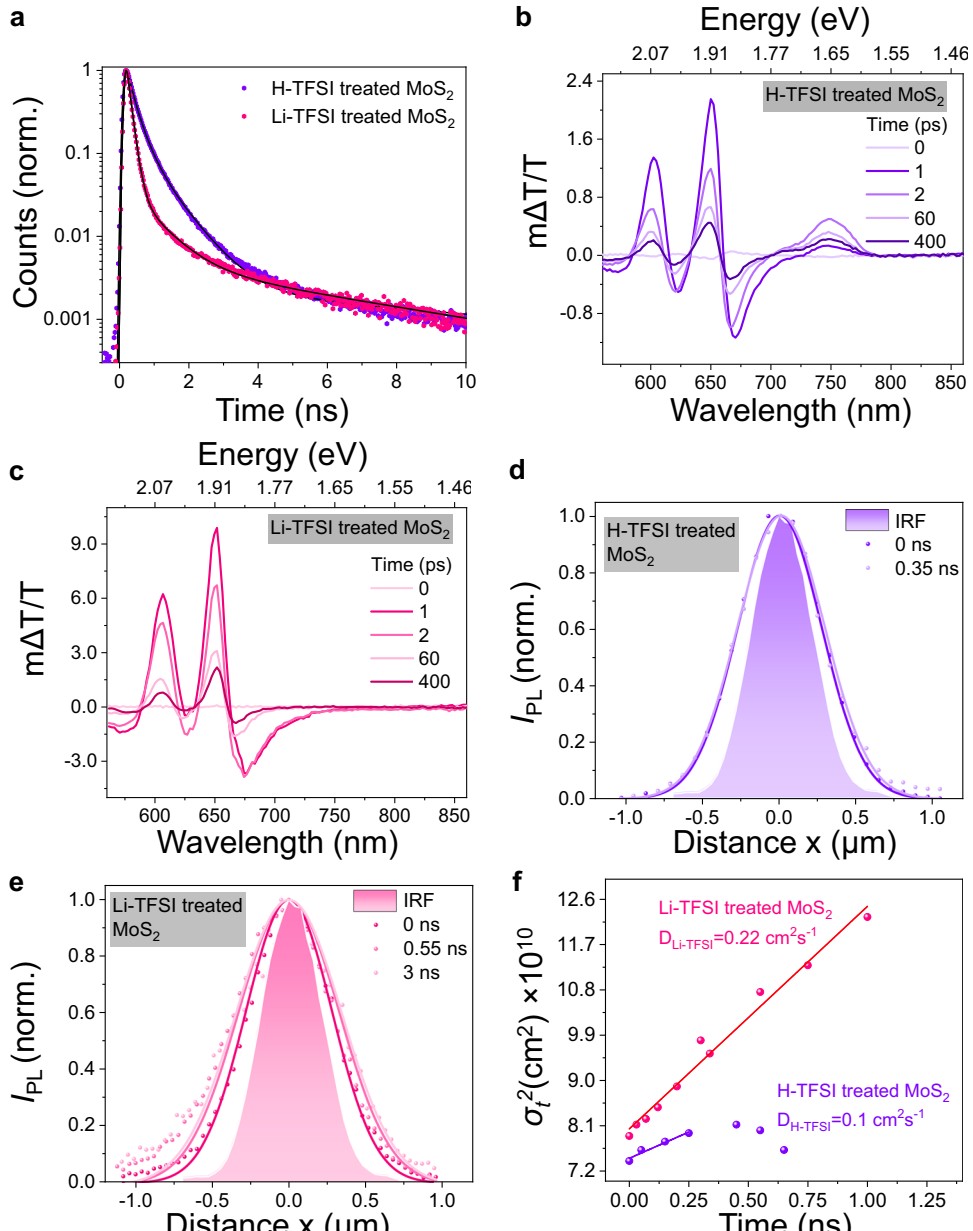

**Fig. 3 Time-resolved photoluminescence (TRPL), pump-probe spectra, and photoluminescence propagation (diffusion) of treated MoS$_2$ monolayers. a** TRPL decay curves for H-TFSI-treated and Li-TFSI-treated monolayer MoS$_2$. The fitting is presented in black solid lines. **b** Pump-probe data of H-TFSI-treated MoS$_2$ where features related to traps can be seen at 730 nm. **c** Pump-probe data of Li-TFSI-treated MoS$_2$ which show no trap-related features at 730 nm. **d** Spatial profile of the normalized PL intensity $I_{PL}$ for time snapshots $t = 0$ and 0.35 ns for H-TFSI-treated monolayer MoS$_2$. IRF refers to the instrument response function. Distance x refers to the distance from excitation. **e** Spatial profile of the normalized PL intensity $I_{PL}$ for time snapshot $t = 0$, 0.55 and 3 ns for Li-TFSI-treated monolayer MoS$_2$. **f** Variance $\sigma_t^2$ as a function of time extracted from the Gaussian PL diffusion profiles of Li-TFSI-treated and H-TFSI-treated MoS$_2$ samples. The diffusion coefficient ($D$) is obtained from fits to the diffusion plots.

blueshifted due to the p-doping effect, and an $A_{2u}$ mode emerges due to Li$^+$ adsorption. In contrast, an $A_{2u}$ mode does not appear in Na-Tf and Li-OAc-treated MoS$_2$ samples (Supplementary Fig. 17b, c), suggesting that the superior PL enhancement effect of Li-TFSI treatment is due to stable adsorption of Li adatom, and low surface charge density as well as non-coordinating nature of TFSI counter anion. The TRPL, PL diffusion, and ultrafast pump-probe measurements were carried out on Li-Tf-treated MoS$_2$ samples to further uncover the role of counter anion play in chemical treatment, as depicted in Supplementary Fig. 18. The normalized average TRPL decay curve is fitted by a three-exponential decay function with $<\tau> \sim 160$ ps showing no

evidence of surface trapping, which is also supported by the pump-probe data. The low $D_{Li-Tf}$ extracted of 0.12 cm$^2$ s$^{-1}$ is, therefore, ascribed to the collision of excitons with excess electrons (trion formation) during the diffusion process[48].

In summary, we have systematically investigated surface chemical treatments that enhance the PL yield of TMDSs by comparing a series of ionic chemicals and small molecule p-dopants, and studying their effect via a range of steady-state and time-resolved spectroscopy and microscopy techniques combined with DFT simulations. Our results provide a detailed mechanistic picture for how these chemical treatments work and allow us to set up selection rules for ionic chemicals to improve PL of TMDSs, where cations and counter

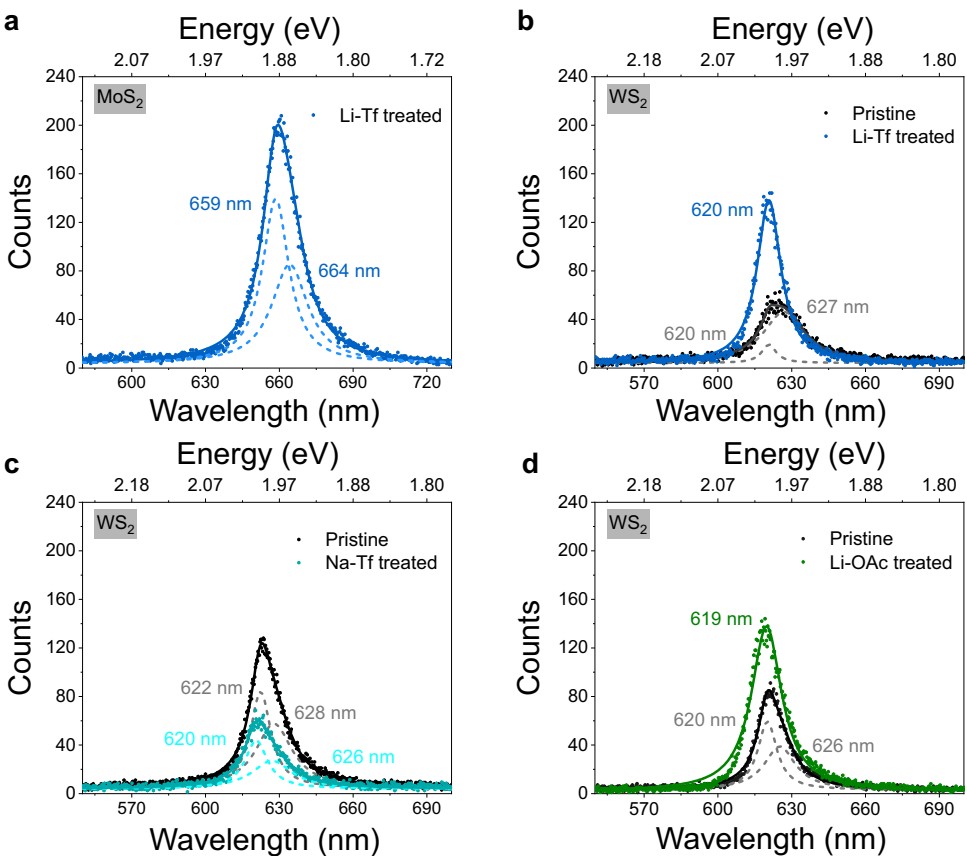

**Fig. 4 Photoluminescence (PL) spectra of M₃-Tf (M₃＝Li and Na) and Li-OAc treated MoS₂ and WS₂ monolayers.** Representative PL spectra for **a** Li-Tf-treated monolayer MoS₂, **b** pristine and Li-Tf-treated monolayer WS₂, **c** pristine and Na-Tf-treated monolayer WS₂, and **d** pristine and Li-OAc-treated monolayer WS₂. The decomposed Lorentzian peak fittings are presented in dashed lines and the cumulative Lorentzian peak fittings are presented in solid lines. Each PL peak position value is stated in the spectra.

anions both play important roles during chemical treatments. The cation must be stably adsorbed on the surface of TMDSs rather than just underdoing electron transfer, allowing for suppression of trion formation, thereby improving PL yield. The counter anion should be non-coordinating with strong electron-withdrawing groups. The strongest enhancement is observed for Li-TFSI, which gives a PL enhancement twice that of the widely discussed "super acid" H-TFSI. More importantly, Li-TFSI is stable and functions in benign solvents, which possesses the potential to be employed directly during device fabrication of TMDSs. Overall, we demonstrate a simple and effective route to enhance PL of TMDSs which opens a route to building high performance chemically treated optoelectronic devices.

## Methods

**Material**. Bulk MoS₂ and WS₂ crystals were purchased from 2D Semiconductors. The monolayer MoS₂ and WS₂ were prepared according to the reported gold-mediated exfoliation method to ensure relatively large monolayers[49]. In this study, all experiments were carried out on monolayers. All chemicals for the surface treatments were purchased from Sigma-Aldrich and used as received.

**Chemical treatments**. The chemical treatments with H-TFSI (0.02 M in 1, 2-dichloroethane), F4TCNQ (0.02 M in dichloromethane), and Magic Blue (0.02 M in dichloromethane) are carried out inside a nitrogen glovebox, and other treatments are carried out in the ambient atmosphere. Methanol is used as a solvent for all ionic salts for comparison. The chemical treatments were achieved by immersing the samples into concentrated solutions of the investigated chemicals (0.02 M) for 40 min.

**Characterization**. The microscope steady-state PL measurement was carried out under ambient conditions using a WITec alpha 300 s setup[50]. Importantly, a 405 nm continuous wave laser (Coherent CUBE) was used as the excitation source. A long-pass filter with a cutoff wavelength of 450 mm was fitted before signal

collection to block excitation scatter. The light was coupled with an optical fiber to the microscope and focused using a 20× Olympus lens. Samples were placed on an X-Y piezo stage of the microscope. The PL signal was collected in reflection mode with the same 20× objective and detected using a Princeton Instruments SP-2300i spectrometer fitted with an Andor iDus 401 CCD detector. The PL maps were measured at 405 nm excitation with a fluence of 15 W cm⁻². The Raman measurements were carried out on a Renishaw inVia Raman confocal microscope with a 532 nm excitation laser in air. The Raman emission was collected by a 20× long working distance objective lens in streamline mode and dispersed by a 1800 l/mm grating with 1% of the laser power (<10 μW). The spectrometer was calibrated to a silicon reference sample prior to the measurement to correct for the instrument response. The XPS measurements were performed using a Thermo Escalab 250Xi system and monochromated aluminum K$_\alpha$ x-ray source. The software package "Thermo Avantage" (Thermo Fisher Scientific Inc., Waltham, USA) was used for data analysis.

The ultrafast pump-probe setup has been described previously[51]. A Light Conversion PHAROS laser system with 400 μJ per pulse at 1030 nm with a repetition rate of 38 kHz is split in two, one part is used to generate the continuum probe light and the second part is used in a Collinear Optical Parametric Amplifier (Orpheus, Light Conversion) to generate the pump source at the desired wavelength. The probe pulse is delayed up to 2 ns with a mechanical delay-stage (Newport). A mechanical chopper (Thorlabs) is used to create an on-off pump-probe pulse series. A silicon line scan camera (JAI SW-2000M-CL-80) fitted onto a visible spectrograph (Andor Solis, Shamrock) is used to record the transmitted probe light. The TRPL microscopy measurements were performed using 405 nm pulsed laser (PDL 828-S "SEPIA II", PicoQuant) excitation via 100× objective in a PicoQuant Microtime 200 confocal setup. The emission signal was separated from the excitation light using a dichroic mirror (Z405RDC, Chroma). The TRPL was measured at 15 μJ cm⁻² and data were averaged from 100 μm² monolayer flakes. PL signals were collected in transmission mode and IRF were measured with blank quartz substrates. For the diffusion measurements, the emission path was raster scanned while the excitation was decoupled and fixed at the center of the sampler ($x = 0$). The PL was then focused onto a Hybrid PMT detector (Picoquant) for single-photon counting (time resolution of 60 ps) through a pinhole (50 μm), with an additional 410-nm longpass filter. Repetition rates of 27 MHz were used for the maps and

the diffusion profiles. The lateral spatial resolution is ~550 nm. An incident power of 60 nW was used, corresponding to a fluence of 700 nJ cm$^{-2}$.

## Data availability

The data that support the findings of this study are available in the University of Cambridge data repository at: https://doi.org/10.17863/CAM.75891. No custom computor code is used in this work.

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

## Acknowledgements

This project has received funding from the European Research Council (ERC) under the European Union's Horizon 2020 research and innovation program (Grant Agreement No. 758826 & 756962). Z.L. acknowledges funding from the Swedish research council, Vetenskapsrådet 2018-06610. S.D.S. acknowledges funding from the Royal Society and Tata Group (UF150033). G.D. acknowledges the Royal Society for funding through a Newton International Fellowship. We acknowledge financial support from the EPSRC and the Winton Programme for the Physics of Sustainability.

## Author contributions

Z.L. designed and conducted the experiments of chemical treatments, Raman and photoluminescence measurements. Z.L. and H.B. carried out the fabrication of monolayer transition metal disulfides. Z.L., H.B. and J.X. analysed the data. Y.Z. performed the computational studies. G.D. performed time-resolved photoluminescence and photoluminescence propagation measurements. A.R. supervised the work. A.L. and S.D.S. contributed to the results interpretation. All authors contributed to the writing of the manuscript.

## Competing interests

The authors declare no competing interests.

## Additional information

**Peer review information** *Nature Communications* thanks Goki Eda and the other anonymous reviewer(s) of this work. Peer reviewer reports are available.

