## [Peer Review File New · Nature Communications]

Reviewers' Comments:

Reviewer #2:

Remarks to the Author:

The authors have addressed my comments with additional data and made their points clearer in the revise version. There are some claims that I still don't fully agree with but the manuscript is now more complete with additional supporting data, and I can recommend publication without further changes (other than some minor typos).

Reviewer #3:

Remarks to the Author:

The authors have addressed all the comments raised by reviewer 1-3 carefully and adequately. They have also added more solid experimental data to support their argument and conclusion. More importantly, they have changed the way of description that cation adsorption indeed contributes to the p doping to enhance the PL, and it is not a totally new enhancement mechanism. The main contribution of this paper is that they find Li-TSFI is a better chemical for PL enhancement (higher enhancement factor, greener), and both cation and counter anion play important roles on the enhancement factor. They have also provide data of many different chemicals on the PL modulation. The reviewer think that this paper provide valuable information for the research community and can be published in Nature Communications without further changes.